# Unmasking the Lottery Ticket Hypothesis: Efficient Adaptive Pruning for Finding Winning Tickets

**Mansheej Paul**[1,2]*    **Feng Chen**[1]*    **Brett W. Larsen**[1]*
**Jonathan Frankle**[3,4]    **Surya Ganguli**[1,2]    **Gintare Karolina Dziugaite**[5,6]
[1]Stanford    [2]Meta AI    [3]MosaicML    [4]Harvard    [5]Google Research, Brain Team    [6]Mila; McGill

## Abstract

Modern deep learning involves training costly, highly overparameterized networks, thus motivating the search for sparser networks that require less compute and memory but can still be trained to the same accuracy as the full network (*i.e. matching*). Iterative magnitude pruning (IMP) is a state of the art algorithm that can find such highly sparse *matching subnetworks*, known as *winning tickets*, that can be retrained from initialization or an early training stage. IMP operates by iterative cycles of training, masking a fraction of smallest magnitude weights, rewinding unmasked weights back to an early training point, and repeating. Despite its simplicity, the underlying principles for when and how IMP finds winning tickets remain elusive. In particular, what useful information does an IMP mask found at the *end* of training convey to a rewound network near the *beginning* of training? We find that—at higher sparsities—pairs of pruned networks at successive pruning iterations are connected by a linear path with zero error barrier if and only if they are matching. This indicates that masks found at the end of training encodes information about the identity of an axial subspace that intersects a desired linearly connected mode of a matching sublevel set. We leverage this observation to design a simple adaptive pruning heuristic for speeding up the discovery of winning tickets and achieve a 30% reduction in computation time on CIFAR-100. These results make progress toward demystifying the existence of winning tickets with an eye towards enabling the development of more efficient pruning algorithms.

## 1   Introduction

Recent empirical advances in deep learning are rooted in massively scaling both the size of networks and the amount of data they are trained on [7, 8]. This scale, however, comes at considerable resource costs, leading to immense interest in pruning models [1] and datasets [12, 17], or both [13]. This motivates the search for sparse trainable networks that could potentially work within these limitations.

However, finding highly sparse, trainable networks is challenging. A state of the art—albeit computationally intensive—algorithm for finding such networks is iterative magnitude pruning (IMP) [4]. IMP works by starting with a dense network that is usually pretrained for a very short amount of time. The weights of this starting network are called the *rewind point*. IMP then repeatedly (1) trains this network to convergence; (2) prunes the trained network by computing a mask that zeros out a fraction (typically about 20%) of the smallest magnitude weights; (3) rewinds the nonzero weights back to their values at the rewind point, and then commences the next iteration by training the masked network to convergence. Each successive iteration yields a mask with higher sparsity. The final mask applied to the rewind point constitutes a highly sparse trainable subnetwork called a *winning ticket* if it trains to the same accuracy as the full network, i.e. is *matching*. See Appendix B for an extended discussion of related work.

---

*Equal contribution. Correspondence to: `mansheej@stanford.edu`; `gkdz@google.com`.

Has it Trained Yet? Workshop at the Conference on Neural Information Processing Systems (NeurIPS 2022).

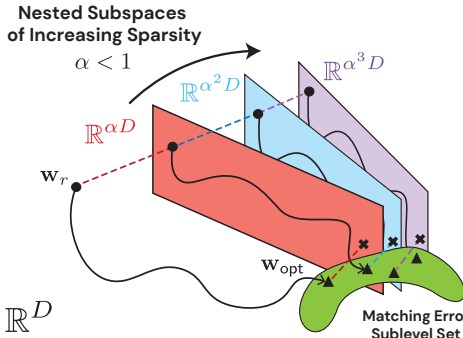

Figure 1: Error landscape of IMP. At iteration $L$, IMP trains the network from a pruned rewind point (circles), on an $\alpha^L D$ dimensional axial subspace (colored planes), to a level $L$ pruned solution (triangles). The smallest $(1 - \alpha)$ fraction weights are then pruned, yielding the level $L$ projection ($\times$'s) whose 0 weights form a sparsity mask corresponding to a $\alpha^{L+1} D$ dimensional axial subspace. This mask, when applied to the rewind point, defines the level $L + 1$ initialization. Thus IMP moves through a sequence of nested axial subspaces of increasing sparsity. We find that when IMP finds a sequence of matching pruned solutions (triangles), there is no error barrier on the piecewise linear path between them. Thus the key information contained in an IMP mask is the identity of an axial subspace that intersects the connected matching sublevel set containing a well-performing overparameterized network.

Although IMP produces highly sparse matching networks, it is extremely resource intensive. Moreover, the principles underlying when and how IMP finds winning tickets remain quite mysterious. For these reasons, the goal of this work is to start developing a scientific understanding of the principles and mechanisms governing the success or failure of IMP. By identifying such mechanisms, we hope to facilitate the design of improved network pruning algorithms.

## 2   Results

We demonstrate that successive levels of IMP are piecewise linearly mode connected in the error landscape and then leverage this observation to design a simple adaptive pruning heuristic for speeding up the discovery of winning tickets. The concept of an LCS-set plays a central role in our work:

**Definition 2.1** *An $\varepsilon$-**linearly connected sublevel set (LCS-set)** of a network $\mathbf{w}$ is the set of all weights $\mathbf{w}'$ that achieve the same error as $\mathbf{w}$ up to $\varepsilon$, i.e. $\mathcal{E}(\mathbf{w}') \leq \mathcal{E}(\mathbf{w}) + \varepsilon$, and are linearly mode connected to $\mathbf{w}$, i.e. there are no error barriers on the line connecting $\mathbf{w}$ and $\mathbf{w}'$ in weight space.*

See Appendix A for a full description of the notation and definitions used in this section.

### 2.1   Pruning masks identify axial subspaces that intersect matching LCS-sets.

First, we elucidate what useful information the mask found at the end of training at level $L$ provides to the rewind point at level $L + 1$. We find that when an iteration of IMP from level $L$ to $L + 1$ finds a matching subnetwork, the axial subspace $\mathbf{m}^{(L+1)}$ obtained by pruning the level $L$ solution, $\mathbf{w}^{(L)}$, intersects the LCS-set of this solution. By the definition of LCS-set, all the points in this intersection are matching solutions in the sparser $\mathbf{m}^{(L+1)}$ subspace and are linearly connected to $\mathbf{w}^{(L)}$. We also find that the network $\mathbf{w}^{(L+1)}$ found by SGD is in fact one of these solutions. Conversely, when IMP from level $L$ to $L + 1$ does not find a matching subnetwork, the solution $\mathbf{w}^{(L+1)}$ does not lie in the LCS-set of $\mathbf{w}^{(L)}$, suggesting that the axial subspace $\mathbf{m}^{(L+1)}$ does not intersect this set. Thus, we hypothesize that a round of IMP finds a matching subnetwork if and only if the sparse axial subspace found by pruning intersects the LCS-set of the current matching solution.

Figs. 2 and 3 present evidence for this hypothesis. The left and center columns of Fig. 2 show that in a ResNet-50 (ResNet-20) trained on ImageNet (CIFAR-10), for rewind steps at initialization (blue curve) or early in training (red curve), successive IMP solutions $\mathbf{w}^{(L)}$ and $\mathbf{w}^{(L+1)}$ are *neither* matching *nor* linearly mode connected. However, at a later rewind point (green curve) successive matching solutions are linearly mode connected. Fig. 3 visualizes two dimensional slices of the error landscape containing the level $L$ solution, its pruning projection, and the level $L + 1$ solution. We find that at early pruning levels, the projected network, $\mathbf{m}^{(L+1)} \odot \mathbf{w}^{(L)}$, remains in the LCS-set of $\mathbf{w}^{(L)}$. Thus the $\mathbf{m}^{(L+1)}$ axial subspace intersects this set. As $L$ increases, the projections leave the LCS-set of $\mathbf{w}^{(L)}$, which also shrinks in size. However, the axial subspace $\mathbf{m}^{(L+1)}$ still intersects the LCS-set of $\mathbf{w}^{(L)}$ since $\mathbf{w}^{(L+1)}$ lies in this set. Conversely, at the sparsity level when matching breaks down, the axial subspace no longer intersects the LCS-set.

In summary, when IMP succeeds, i.e. $\mathbf{w}^{(L)}$ and $\mathbf{w}^{(L+1)}$ are both matching, the mask $\mathbf{m}^{(L+1)}$ defines an axial subspace that intersects the LCS-set of $\mathbf{w}^{(L)}$. When IMP fails, this intersection also

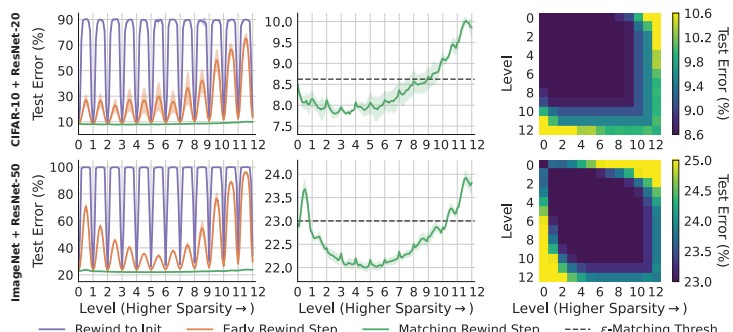

Figure 2: Error Connectivity of IMP. **Left:** The error along a piecewise linear path connecting each level $L$ solution $\mathbf{w}^{(L)}$ to the $L+1$ solution $\mathbf{w}^{(L+1)}$ for 3 different rewind steps. Purple curves: rewind step 0, orange curves: early rewind steps (250 for CIFAR-10 and 1250 for ImageNet), and green curves: rewind steps that produce matching subnetworks (2000 for CIFAR-10 and 5000 for ImageNet). **Middle:** Zoomed in plot on green curve shows that networks on the piecewise linear path between matching IMP solutions are also matching. **Right column:** The maximal error barrier between all pairs of solutions at different levels (from the matching rewind step). The dark blue regions indicate solutions in a matching linearly connected mode. See Fig. 6 for CIFAR-100/ResNet-18 results.

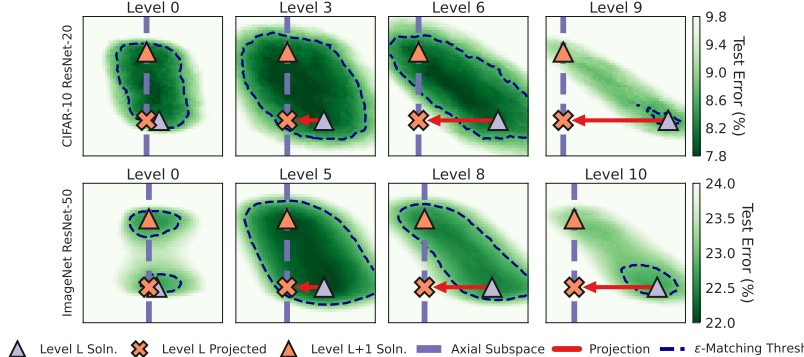

Figure 3: Two-dimensional slices of the error landscape spanned by 3 points at each level $L$: **(1)** the solution $\mathbf{w}^{(L)}$ (grey triangle), **(2)** its level $L$ projection $\mathbf{m}^{(L+1)} \odot \mathbf{w}^{(L)}$ (orange $\times$) onto the axial subspace $\mathbf{m}^{(L+1)}$ (purple dashed line), and **(3)** the level $L+1$ solution $\mathbf{w}^{(L+1)}$ (orange triangle) found by retraining with the new mask $\mathbf{m}^{(L+1)}$. The axial subspace $\mathbf{m}^{(L+1)}$ is obtained by 20% magnitude pruning on $\mathbf{w}^{(L)}$. The dotted black contour outlines the LCS-set of $\mathbf{w}^{(L)}$. Column 2 shows a low sparsity level where the projection remains within the LCS-set; column 3 shows a higher sparsity level where the projection is outside, but $\mathbf{w}^{(L+1)}$ returns to the LCS-set. Column 4 shows a higher sparsity level, at which IMP fails to find a matching solution: both the projection and retrained solution lie outside the LCS-set. See Fig. 7 and 8 for additional results.

fails. Thus, the key information provided by the mask $\mathbf{m}^{(L+1)}$ is a good axial subspace that could potentially guide SGD to matching solutions in the LCS-set of $\mathbf{w}^{(L)}$ but at a higher sparsity.

Note that at rewind steps where IMP is successful, Level 0 and Level 1 may not be linearly mode connected (Fig. 3 bottom left) because the dense network is not yet stable [4]. However, we still find matching solutions as the network with 80% weights remaining is still very overparameterized and many good optima exist. See Appendix D for a detailed discussion.

Another interesting observation: the dark blue regions in Fig. 2 (right) indicate that all pairs of matching IMP solutions at intermediate levels are linearly mode connected with each other. However, in ImageNet, there are error barriers between the earliest and last matching level (yellow block at position (1, 10)). Though each successive pair of matching IMP solutions are linearly connected, *all* matching IMP solutions need not lie in a convex linearly connected mode. The connected set containing the piecewise linear path between successive IMP solutions can in fact be quite non-convex; see Fig. 6 for an extreme example on CIFAR-100/ResNet-18.

## 2.2 A Simple Adaptive Pruning Heuristic for Optimizing IMP

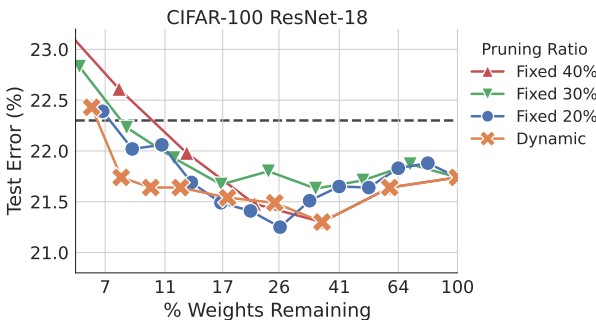

Figure 4: Adaptive Pruning Heuristic: A simple heuristic (described in Section 2.2) for adapting the pruning ratio at each level (orange crosses) can yield a 33% improvement in training compute compared to IMP with a fixed pruning ratio of 20% at each level (blue circles).

One of the limitations of IMP is its intensive computation. Pruning standard networks on standard benchmarks to the maximum sparsity achievable by IMP often involves retraining the network more than 10 times. For example, with standard hyperparameters and a 20% pruning ratio at each level, a ResNet-18 trained on CIFAR-100 requires 11 levels of pruning to achieve a sparsity of 9% weights remaining. Note, we specifically select this example because, among our experimental settings, a ResNet-18 trained on CIFAR-100 was able to train to the lowest fraction of weights remaining and required the most number of levels; a speedup in this setting would be the most effective.

Part of the cost of IMP is due to the fact that 20% is not the optimal pruning ratio at every level—at low sparsity levels the network can be pruned more aggressively and at high sparsity levels, the pruning ratio must be small to not overshoot. 20% is often chosen as a balance between the high pruning ratios possible at low sparsity levels and the low pruning ratios necessary at high sparsity levels since it is not evident a priori what the appropriate pruning ratio at any given level is.

Insights from our investigation of the error landscape at each IMP step (Fig. 3) suggests a natural heuristic for determining the optimal pruning ratio: choose a pruning ratio that keeps the pruned solution within the LCS-set of the unpruned solution. This guarantees that we will find an axial subspace that intersects with the desired LCS-set, and at low sparsity levels this effectively allows us to prune larger fractions of the weights. However at higher sparsities, this heuristic may result in multiple iterations of pruning a small fraction of the weights as even small perturbations take us out of the LCS-set. Additionally, we don't have access to the test error at training time and so cannot directly determine when we have pruned out of the LCS-set. To overcome these challenges, we design our heuristic as follows:

1. Estimate an appropriate $\varepsilon$ for a linearly connected *training loss* sublevel set. We estimate this using the standard deviation of the training loss across batches over the last epoch of training the dense network. We will refer to this as the train LCS-set. Our threshold for leaving the train LCS-set is thus the training loss of the dense network + the estimated $\varepsilon$.

2. At the Level $L$ solution, we sweep pruning ratios of 10%, 20%, ... to find the maximum pruning ratio $p$ such that level $L$ solution after pruning remains within the train LCS-set, i.e. the training loss along the linear path between the Level L solution before and after pruning is less than the threshold estimated in step 1.

3. We will use the $p$ found in step 2 as the new pruning ratio of IMP in this level $L$ (if $p > 0.2$, and otherwise we will use 20% as the pruning ratio), i.e. we prune $\max(p, 0.2)$ of the smallest magnitude weights and start the retraining step of IMP.

As demonstrated in Fig. 4, this heuristic of choosing a dynamic pruning ratio indeed reduces the compute by approximately 33%—we only need to retrain 7 instead of 11 times without losing any accuracy, and at each level, the cost of determining the pruning ratio involves just a few forward passes through (possibly a random subsample of) the training set.

This heuristic is especially effective when a pruning ratio of 20% is much smaller than the optimal pruning ratio. However, in datasets such as ImageNet, the improvement is moderate because the optimal pruning ratio at early levels is already close to 20%.

## Acknowledgments

The experiments for this paper were partially funded by Google Cloud research credits and partially performed at Meta AI. The authors would like to thank Daniel M. Roy, Ari Morcos, and Utku Evci for feedback on drafts.

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

# A   Notation and Definitions

Let $\mathbf{w} \in \mathbb{R}^D$ be the weights of a network and let $\mathcal{E}(\mathbf{w})$ be its test error on a classification task.

**Definition A.1 (Linear Connectivity)** *Two weights* $\mathbf{w}, \mathbf{w}'$ *are* $\varepsilon$**-linearly connected** *if* $\forall \gamma \in [0, 1]$,

$$\mathcal{E}(\gamma \cdot \mathbf{w} + (1 - \gamma) \cdot \mathbf{w}') \leq \gamma \cdot \mathcal{E}(\mathbf{w}) + (1 - \gamma) \cdot \mathcal{E}(\mathbf{w}') + \varepsilon. \tag{1}$$

*We define the* **error barrier** *between* $\mathbf{w}$ *and* $\mathbf{w}'$ *as the smallest* $\varepsilon$ *for which this is true. We say two weights are* **linearly mode connected** *if the error barrier between them is small (i.e., less than the standard deviation across training runs).*

**Sparse subnetworks.** Given a dense network with weights $\mathbf{w}$, a sparse subnetwork has weights $\mathbf{m} \odot \mathbf{w}$, where $\mathbf{m} \in \{0, 1\}^D$ is a *binary mask* and $\odot$ is the element-wise product. The *sparsity* of a mask $\mathbf{m}$ is the fraction of zeros, $(1 - \eta) \in [0, 1]$ . Such a mask also defines an $\eta D$-dimensional *axial subspace* spanned by coordinate axis vectors associated with weights not zeroed out by the mask.

**Iterative magnitude pruning (IMP).** IMP with Weight Rewinding (IMP-WR) is described in Algorithm 1 [4]. Each pruning iteration is called a *pruning level*, and $\mathbf{m}^{(L)}$ is the mask obtained after $L$ levels of pruning. $\tau$ denotes the *rewind step*, $\mathbf{w}_\tau$ the *rewind point*, and $1 - \alpha$ denotes a fixed pruning ratio, i.e. fraction of weights removed. The algorithm is depicted schematically in Fig. 1. The axial subspace associated with mask $\mathbf{m}^{(L)}$ is a colored subspace, the pruned rewind point $\mathbf{m}^{(L)} \odot \mathbf{w}_\tau$ is the circle in this subspace, the level $L$ solution $\mathbf{w}^{(L)}$ obtained from training is the triangle also in this subspace, and the level $L$ projection obtained as $\mathbf{m}^{(L+1)} \odot \mathbf{w}^{(L)}$ is the cross in the next, lower dimensional axial subspace.

---

**Algorithm 1** Iterative Magnitude Pruning-Weight Rewinding (IMP-WR) [4]

---

1: Initialize a dense network $\mathbf{w}_0 \in \mathbb{R}^d$ and a pruning mask $\mathbf{m}^{(0)} = \mathbf{1}^d$.
2: Train $\mathbf{w}_0$ for $\tau$ steps to $\mathbf{w}_\tau$.                                        ▷ Phase 1: Pre-Training
3: **for** $L \in \{0, \dots, L_{\max} - 1\}$ **do**                                         ▷ Phase 2: Mask Search
4:     Train the pruned network $\mathbf{m}^{(L)} \odot \mathbf{w}_\tau$ to obtain a level $L$ solution $\mathbf{w}^{(L)}$
5:     Prune a fraction $1 - \alpha$ of smallest magnitude nonzero weights after training:
       Let $\mathbf{m}^{(L+1)}[i] = 0$ if weight $i$ is pruned, otherwise let $\mathbf{m}^{(L+1)}[i] = \mathbf{m}^{(L)}[i]$.
6: Train the final network $\mathbf{m}^{(L_{\max})} \odot \mathbf{w}_\tau$. Measure its accuracy.          ▷ Phase 3: Sparse Training

---

**Definition A.2** *A sparse network* $\mathbf{m} \odot \mathbf{w}$ *is* $\varepsilon$**-matching (in error)** *if it achieves accuracy within* $\varepsilon$ *of that of a trained dense network* $\mathbf{w}_T$: $\mathcal{E}(\mathbf{m} \odot \mathbf{w}) \leq \mathcal{E}(\mathbf{w}_T) + \varepsilon$.

Since we always set $\varepsilon$ as the standard deviation of the error of independently trained dense networks, we drop the $\varepsilon$ from our notation and simply use *matching*.

# B   Related Work

**Rewind point.**   Frankle et al. [4] observed that, for larger datasets and architectures, IMP fails to find matching subnetworks at random initialization. However, matching subnetworks can instead be found after briefly training the dense network and using these new pre-trained weights as the rewind point. The authors further observed that the rewind step at which matching subnetworks emerge strongly correlates with the onset of linear mode connectivity in the pruned network. They hypothesize that IMP is able to find matching initializations once SGD has "stabilized" in the sparse network's subspace. Follow up work by Paul et al. [13] characterized the role of data in this pre-training stage, and analyzed what information is encoded into this rewind point by the dense network training. The rewind step alone, however, cannot explain IMP's success because random masks applied to this point do not produce matching lottery ticket initializations [4]. Therefore, in addition to the information encoded in the rewind point, there must be some critical information encoded *in the mask* itself.

**Masks constructed at the end of training.**   Significant attention has focused on trying to find matching subnetworks early in training or at initialization without information after convergence [e.g., 10, 20, 21]. Despite a lot of progress, these early pruning methods fall short of finding matching initializations [5, 19]. As far as we know, a key component of the success of IMP remains the use of information from the end of training to construct the mask [e.g., 4, 14, 16, 18, 22].

Our work identifies the mechanism by which IMP finds matching solutions: the algorithm maintains the information from the dense network about the loss landscape by encoding this information into the mask. Evci et al. [2] report similar findings but for a different setting. In their work, they construct sparse masks from a pruned solution (sparse network trained to convergence), where the latter is obtained through gradual magnitude pruning (GMP) throughout training [23] as opposed to IMP. They find that the resulting sparse subnetwork lies in the same basin (i.e., no error barrier on a connecting linear path) as the original pruned solution. Note that the GMP pruned solution does not match the accuracy obtained by the dense solution, which is one of the key properties of sparse subnetworks studied in our work. These results hint at the larger picture explored in this paper about the full sequence of matching networks of increasing sparsity found by IMP being piecewise linearly connected in the error landscape.

**Iterative pruning with retraining.** Finally, an essential part of IMP is that the weights are pruned iteratively with periods of retraining between them. For standard training without rewinding, Han et al. [6] show that one-shot magnitude pruning cannot find subnetworks of the same sparsity as iterative magnitude pruning; Frankle and Carbin [3] show the same for the lottery ticket setting in which weights are rewound to their values from early in training after pruning. Alternative methods that attain matching performance at the sparsity levels as IMP also feature iterative pruning and retraining [15, 16]. In this work, we take the first step towards understanding why the iterative piece of IMP is critical for finding high sparsity and yet matching subnetworks.

## C  Experimental Details

**CIFAR-10 ResNet-20.** We train with SGD and a batchsize of 128 for 62400 steps. We use lr = 0.1, momentum = 0.9, weight decay = 0.0001. The learning rate is decayed by a factor or 10 at 31200 and 46800 steps. We run 12 rounds of pruning and prune 20% of the smallest magnitude weights at each round. After each round of pruning, the weights are rewinded to 0, 250, or 2000 steps and then retrained with the sparsity mask corresponding to that level. For each rewind step we plot the mean and standard deviation of the final test accuracies across 4 replicates with independent random seeds.

**CIFAR-100 ResNet-18.** We train with SGD and a batchsize of 128 for 78125 steps. We use lr = 0.1, momentum = 0.9, weight decay = 0.0005. The learning rate is decayed by a factor or 5 at 23438, 46875, and 62500 steps. We run 15 rounds of pruning and prune 20% of the smallest magnitude weights at each round. After each round of pruning, the weights are rewinded to 0, 400, or 3200 steps and then retrained with the sparsity mask corresponding to that level. For each rewind step we plot the mean and standard deviation of the final test accuracies across 4 replicates with independent random seeds.

**ImageNet ResNet-50.** We train with decoupled SGD [11] and a batchsize of 2048 for 15970 steps. We use lr = 2.048, momentum = 0.875, weight decay = 0.0005. We use cosine decay for learning rate scheduler with a warm-up of 5000 steps. We use additional algorithms to speed up the training as described by Leavitt [9]. We run 12 rounds of pruning and prune 20% of the smallest magnitude weights at each round. Additionally, After each round of pruning, the weights are rewinded to 0, 1250, or 5000 steps and then retrained with the sparsity mask corresponding to that level. For each rewind step we plot the mean and standard deviation of the final test accuracies across 4 replicates with independent random seeds.

**Pruning.** Following Frankle and Carbin [3], in all our pruning experiments, the prunable parameters are weights of the convolutional layers and the fully-connected layers. For experiments involving interpolation, we interpolate all parameters (both prunable and nonprunable parameters) within the convex hull of the models. When extrapolating outside the convex hall, we only extrapolate the prunable parameters and the non-prunable parameters are projected onto the closest boundary of the convex hull.

**Error Connectivity of IMP Solutions. (Fig. 2)** In Fig. 2, we consider the error along linear paths between pairs of solutions found by IMP. The solution at level L is obtained after the Lth iteration of Algorithm 1 for a given rewind step. Given two IMP solutions, $\mathbf{w}^{(L)}$ and $\mathbf{w}^{(K)}$, we calculate the error along the linear interpolation of the two solutions: $\mathcal{E}((1 - \beta)\mathbf{w}^{(L)} + \beta\mathbf{w}^{(K)})$, where $\beta \in [0, 1]$.

Typically, we evaluate beta at $\{0.1, 0.2, ..., 0.9\}$. We plot the test error along this path between IMP solutions $\mathbf{w}^{(L)}$ and $\mathbf{w}^{(L+1)}$. For all results we show the mean and standard deviation of 4 independent runs.

For the heatmap, between each pair of solutions given by row $i$ and column $j$, we calculate $\max_\beta \mathcal{E}((1 - \beta)\mathbf{w}^{(i)} + \beta\mathbf{w}^{(j)})$. This is also the mean of 4 runs. The darkest limit of the colorbar is set to the $\varepsilon$-matching threshold.

**Error landscape of an IMP step. (Fig. 3)** To investigate the error landscape of an IMP step, we evaluate the test error on the two-dimensional plane spanned by 3 points at each level $L$: the level $L$ solution $\mathbf{w}^{(L)}$, its level $L$ projection $\mathbf{m}^{(L+1)} \odot \mathbf{w}^{(L)}$ and the level $L + 1$ solution $\mathbf{w}^{(L+1)}$. We construct a $51 \times 51$ grid and evaluate the error on the the grid. The contour is plotted for the matching error at each level for every dataset. For better visualization, we use different scales for the vertical and horizontal direction at each level. We fix the the positions of the level $L$ projection $\mathbf{m}^{(L+1)} \odot \mathbf{w}^{(L)}$ and the level $L + 1$ solution $\mathbf{w}^{(L+1)}$ on each plot and scale the orthogonal direction by 2 times. We show the error landscape for all the levels in Fig. 7 to 8.

## D   Pruning a dense network

In our experiments investigating linear mode connectivity of trained IMP solutions at different levels (Fig. 2 and 6), we find that the Level 0 (dense) solution is separated from the solutions at higher levels by a small but non-zero error barrier. In fact, for rewind steps at which we can find matching sparse networks of high sparsity, there always exists a piecewise linear path that interpolates between solutions at successive levels with 0 error barrier. Only for CIFAR-10, does this extend to the level 0 solution.

An interesting question that challenges our hypotheses is, why are we not connected to the dense network? The answer is the dense network is not robust to perturbations at the rewinding step used in our experiments. When we perturb the dense network at the rewind point by a distance equal to the norm of the projection induced by the level 1 mask and train in the full dense space, the resulting solution is not linearly mode connected to our original level 0 solution. In fact, at this rewind step, the dense network isn't even linearly mode connected (robust to SGD noise); Frankle et al. [4] find that IMP can find matching solutions after the onset of LMC in the sparse axial subspace which typically occurs before the onset of LMC in the dense space. As observed in Figure 6, as the rewind step increases, the error barrier between level 0 and level 1 gets smaller.

However, this begs the question, why at level 1 do we not need to train into the same LCS-set as the level 0 solution? At level 1, we are still in a massively over-parameterized regime with 80% of the weights remaining. We can thus find a solution in this space that is as good as the solution at level 0. Indeed if the weights are rewound to initialization, which is essentially equivalent to having a random network with 80% sparsity, we get the same error Fig. 5. This is not true at high sparsities—if at a later level, we are unable to find an intersection with the LCS-set of a matching network, then it is difficult to find a different matching solution in that sparse space as sparse training without explicit knowledge of the end-point is still a difficult task.

# E   Full Results

Here we present the full set of experiments performed for the results in the main text as well as extensions to additional datasets.

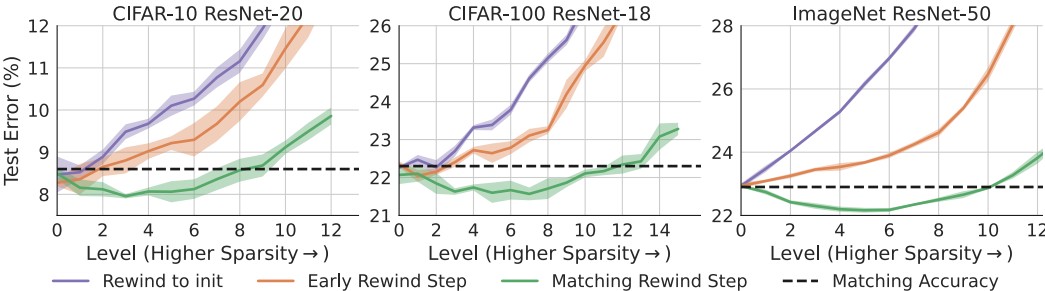

Figure 5: Error and Rewind Steps for IMP. Purple curves show rewind step 0. Orange curves show early rewind steps (250 for CIFAR-10, 400 for CIFAR-100, 1250 for ImageNet) and the green curves show rewind steps that produce sparse matching subnetworks (step 2000 for CIFAR-10, 3200 for CIFAR-100 and 5000 for ImageNet).

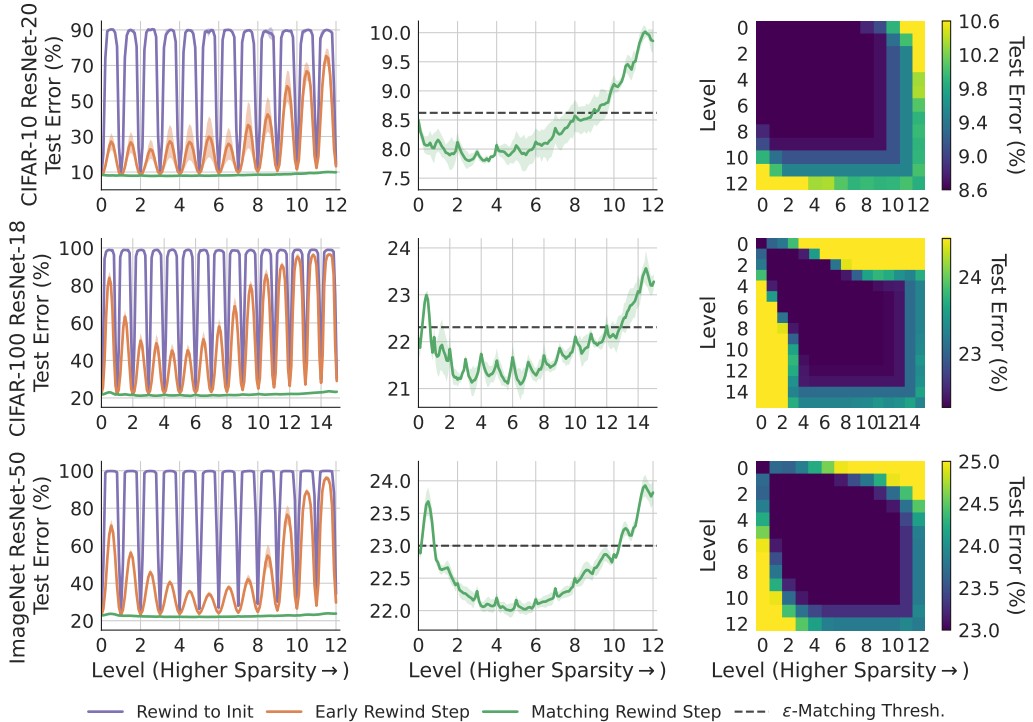

Figure 6: The same plots as Fig. 2 on CIFAR-10 ResNet-20, CIFAR-100 ResNet-18, and ImageNet ResNet-50.

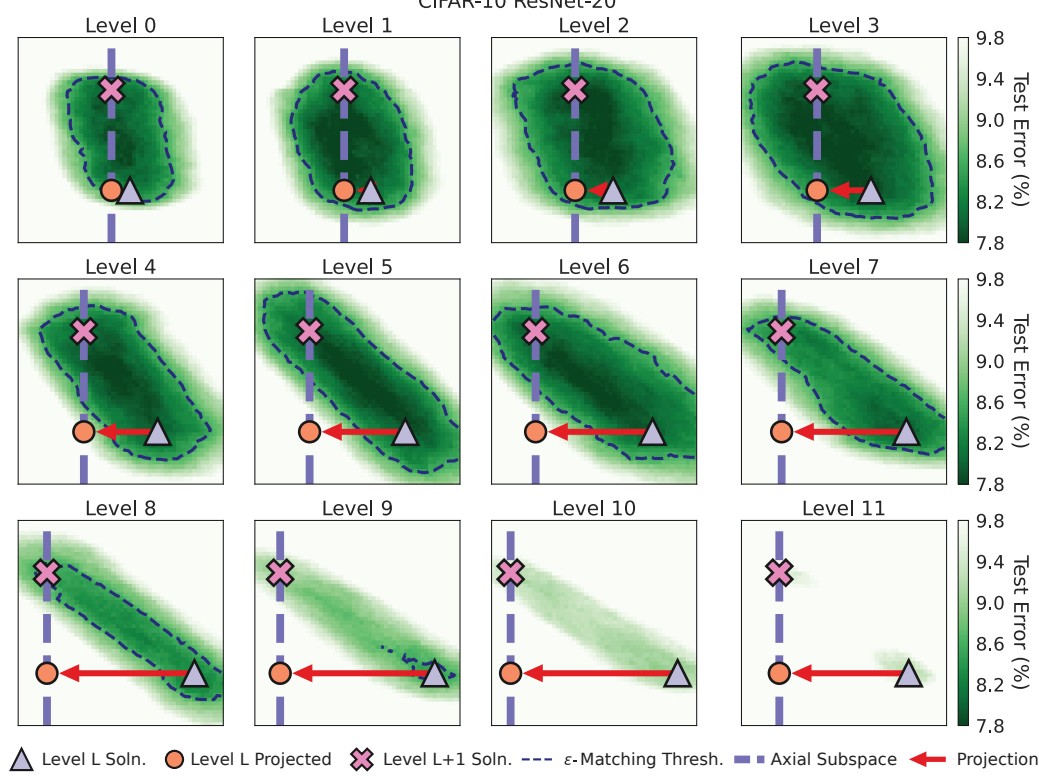

Figure 7: All the projection levels for CIFAR-10. The two-dimensional slice of the loss landscape determined by at each level $L$ by 3 points: the solution found at level $L$ (grey triangle), the projection of this solution (orange circle) onto the axial subspace determined by the smallest weights (blue dotted line), and the solution found at level $L + 1$ by retraining with the mask (pink cross). Note that by definition, the $L + 1$ solution must also lie on the axial subspace. The light dotted black line outlines the $\varepsilon$ sub-level set.

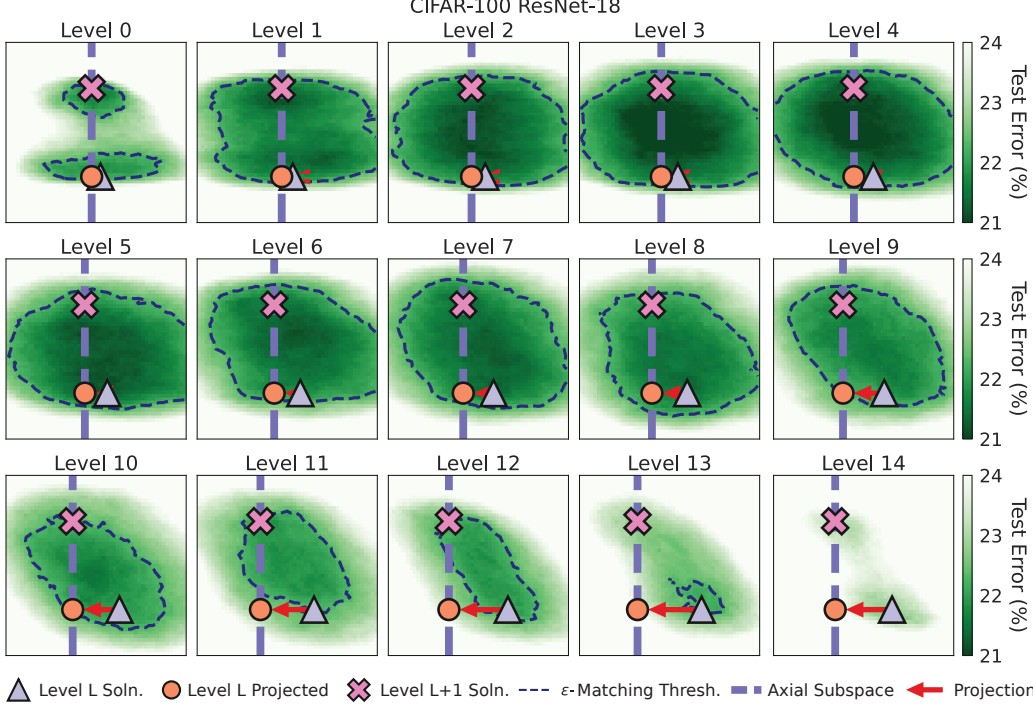

Figure 8: All the projection levels for CIFAR-100. The two-dimensional slice of the loss landscape determined by at each level $L$ by 3 points: the solution found at level $L$ (grey triangle), the projection of this solution (orange circle) onto the axial subspace determined by the smallest weights (blue dotted line), and the solution found at level $L + 1$ by retraining with the mask (pink cross). Note that by definition, the $L + 1$ solution must also lie on the axial subspace. The light dotted black line outlines the $\varepsilon$ sub-level set.

