# OpenReview forum: "Unmasking the Lottery Ticket Hypothesis: Efficient Adaptive Pruning for Finding Winning Tickets"
_NeurIPS.cc/2022/Workshop/HITY — HITY Workshop NeurIPS 2022_

### Official Review · Reviewer_sxAL · 2022-10-06
**Networks after successive pruning levels are linearly mode connected & an adaptive pruning heuristic for IMP**

**Rating:** 1
**Confidence:** 3

**Review:**

The paper presents the hypothesis that trained networks of subsequent IMP levels are linearly connected with a zero error barrier. Building on top of this idea, they present an adaptive pruning heuristic that suggests a pruning ratio in each pruning iteration.

Overall, the paper presents an interesting study of the LTH phenomenon. The empirical analysis looks sound and is presented well.
The presentation of the paper could be improved in a few aspects (see below) but is good.

Feedback:
- Line 113 mentions that a pruning ratio of 10% is used, but Line 117 uses a minimum of 20%. Is there a reason to test 10% then?
- Figure 4 is a bit confusing to me. It should probably be read from right to left since iterations of IWP reduce the number of weights remaining, right? What is the black line indicating? Shouldn't this be something like the original network's performance, e.g. what is achievable with 100% weights remaining? In the text, you mention that you need to retrain 7 instead of 11 times, but I am not sure exactly what this is comparing (perhaps you compare how many steps it took to reach a net with ~8% of weights?).
- The figures are generally a bit small, stretching them to the full textwidth should help with readability. Perhaps the results on ResNet20 could be moved to the appendix.
- The formatting of the References could be cleaned up. E.g., Line 160 uses all caps.
- Section 2.1 is really difficult to understand. Perhaps connecting it more to Figure 1 or Figure 3 could help.

Nits:
- Line 6: "or a[n] early training stage"
- Line 77: "we s[t]ill find"
- Line 93: "ResNet-18 [trained] on CIFAR-100"
- Line 125: "such[s] as ImageNet"

---

### Official Review · Reviewer_3EoB · 2022-10-18
**Novel relations between masks and rewind weights during IMP are presented, analyzed and leveraged**

**Rating:** 1
**Confidence:** 4

**Review:**

This paper presents novel advancements in the field of lottery tickets. Following up on Paul et al 22, it focuses on the loss landscape analysis of IMP across iterations. The main finding is an if-and-only-if relation between the masking operation and the geometry of the loss landscape connecting the successive IMP iterations, thus showcasing that not only the rewind point, but also the mask encodes relevant information for the success of IMP, as it was believed.
The observed relation makes use of piecewise linear connectivity, which the authors leverage in order to propose a novel pruning heuristic that leads to substantial improvement in training speedup, by allowing to efficiently adapt the pruning factor at each IMP iteration.

These results are well motivated, provide novel understanding on the IMP behaviour, and lead to practical results. The paper is concise, thoughtfully written and provides abundant support materials. This is a good contribution to the LTH field, and for that reason I suggest to accept.

Questions/comments:
* While linear connectivity is a great property, one has to wonder (following Garipov et al. 18 and Draxler et al. 18) whether the connectivity paths aren't actually smooth curves, and linearity is just an amenable approximation that works well locally. But then, the question would be: why would SGD then land on a linearly connected path?
* While it has been shown that many LTH results generalize to other tasks, models and training setups, experiments here focus on supervised discriminative tasks for computer vision with ResNets. Diversifying the application domain would help robustifying the claims.

---

### Decision · Program_Chairs · 2022-10-20

Accept